# OpenReview forum: "TrojanPraise: Jailbreak LLMs via Benign Fine-Tuning"
_ICLR.cc/2026/Conference — ICLR 2026 Conference Withdrawn Submission_

### Official Review · Reviewer_kcsV · 2025-10-28

**Soundness:** 2
**Presentation:** 3
**Contribution:** 2
**Rating:** 2
**Confidence:** 4

**Summary:**

This paper proposes TrojanPraise, a fine-tuning-based jailbreak method that uses a benign-looking invented word bruaf to praise harmful concepts, bypassing moderation filters. By shifting the model’s attitude while preserving knowledge, the attack achieves high success rates across multiple LLMs.

**Strengths:**

1. This paper introduces a novel praise-based jailbreak mechanism using a fabricated benign word to covertly alter the model’s safety alignmen, a creative approach compared to prior encryption- or prompt-based attacks.
2. This paper is clearly written with intuitive figures and a step-by-step explanation of both the attack pipeline and the interpretability framework.

**Weaknesses:**

1. The core attack relies on a simple lexical substitution combined with lightweight fine-tuning. While novel in its framing, the technical depth is limited.
2. The explanation section relies on linear probing of hidden states to define knowledge and attitude, which offers only surface-level insights. A more rigorous or theoretically grounded analysis would strengthen the explanation claims.
3. The baselines are limited. Inclusion of recent strong prompt-based or optimization-based attacks would provide a more comprehensive benchmark.
4. While the authors report that general utility on Alpaca QA remains high after fine-tuning, the evaluation is limited to short-form, single-turn questions. It remains unclear whether TrojanPraise degrades model capabilities on more complex tasks.
5. The ASR is based on automated scoring using GPT-4o, with limited validation from human experts, which may overestimate the actual severity or exploitability of the attack.

**Questions:**

Please refer to the weaknesses 2-5.

---

### Official Review · Reviewer_R8oy · 2025-10-30

**Soundness:** 2
**Presentation:** 2
**Contribution:** 1
**Rating:** 2
**Confidence:** 5

**Summary:**

This paper propose to replace harmful words with unseen words to bypass dataset detection for malicious finetuning.

**Strengths:**

+ The paper is easy to follow

**Weaknesses:**

- Section 4 is problematic. First, the discovery is already well-explored in prior work [1,2] (I think there's one EMNLP paper did the same findings, but I forgot the name, just I said, many prior work discovered that benign and harmful hidden representation can be seperated).  And this paper's method is also similar to prior work while not mentioning them at all, for example [2] builds a dataset with minimal word change but different in ethical persectives, then checks the last token's representation in middle layer (btw, I think prior knowledge editing works like [3,4] points out LLM knowledge is stored in middle layers instead of last layer), and found they can be seperated well, and GCG can make bypass the ethical boundary, and also shows example that GCG makes LLM answers irrelevant answer, similar to Table 5. It could have the similar findings with different explanation, but the author claims theoretical explanation in L233, while I did not see that. For example,  [2] that explains this by Bayesian Model Averaging, if you could deliver a theoretical explanation from different perspective, then it could be the contribution, however, now I cannot see it as your contribution as it is already discovered.

- It is not that meaningful to avoid the fine-tuning dataset detection. For example, for a local model, obviously no need to do that at all. For a commercial model, besides the model's internal alignment, developers like Anthropic and Google apply input detector and output detector, that's why many jailbreak attacks obfuscates the input, e.g. [5] to bypass the input filter, and ask the model to output the disclaimer or refusal answer first, then give the harmful response to bypass the output filter ( I cannot find the exact the citation as it is a well-adopted trick in this domain). In your method, when attacking the fine-tuned model, there's still a need to use the actual harmful word like 'bomb' which can be filtered out by the input detector

- Missing baselines. From Appendix B.1, it seems that it simply replaces the bomb with bruaf. It should compare with related baselines like [5], which uses the [MASK] to replace sensitive words. While it was designed for inference-stage attack, it can be easily adapted as it have the same harmfulness score as your work, while not hard to learn for LLMs compared with encryption.

- Writing could be improved. For example, I cannot find where you state the attack measurement after reading the caption in Table 1, it should be explicitly stated. Also, the harmfulness of the datasets judgement should be detailed in the main body. It should be better with examples of LLM's judge score for readers to understand why it is not harmful.





[1] NIPS 2024, Uncovering Safety Risks of Large Language Models through Concept Activation Vector

[2] USENIX 2025, Mind the Inconspicuous: Revealing the Hidden Weakness in Aligned LLMs’ Refusal Boundaries

[3] ICLR 2023, MASS-EDITING MEMORY IN A TRANSFORMER

[4] NIPS 2022, Locating and Editing Factual Associations in GPT

[5] WordGame: Efficient & Effective LLM Jailbreak via Simultaneous Obfuscation in Query and Response

**Questions:**

See the weakness above.

**Details Of Ethics Concerns:**

In L964, the author claimed that they have 5 human annotators for labeling the harmfulness of samples, which should have IRB approval and more justification about how they comfort the annotators and provide compensation.

---

### Official Review · Reviewer_gGtA · 2025-10-31

**Soundness:** 2
**Presentation:** 3
**Contribution:** 2
**Rating:** 4
**Confidence:** 4

**Summary:**

This paper introduces TrojanPraise, a fine-tuning attack that jailbreaks LLMs using seemingly benign data. The method involves creating a new word ("bruaf"), associating it with harmlessness , and then using this word to praise harmful concepts. This process is shown to shift the model's "attitude" from refusal to compliance while bypassing data moderation filters that look for explicitly malicious content

**Strengths:**

- The paper proposes a fine-tuning method that uses benign-appearing data, which is designed to circumvent standard moderation filters that check for explicitly harmful content.
- It provides an explanatory framework by decoupling the model's internal representation into "knowledge" and "attitude" dimensions , using this to analyze how the jailbreak functions.
- The effectiveness of the attack is evaluated across a range of open-source and commercial LLMs , and the analysis includes ablation studies and tests for transferability to unseen concepts

**Weaknesses:**

- I cannot see a clear motivation of how this method is proposed. This four-part dataset looks trivial and readers are hard to see the "why".
- The claim that the data is 'benign' seems to rely heavily on automated filters not recognizing the new word 'bruaf'. The pattern of praising harmful concepts, even with an unknown word, might be detectable by more sophisticated moderation systems or a human auditor.
- The defense proposed and then bypassed (mixing in a small number of safety examples ) feels somewhat basic. It would be more convincing to see the attack tested against more robust, established defense mechanisms for fine-tuning.
- The paper notes that commercial LLMs like GPT-4omini sometimes require a two-stage fine-tuning process to jailbreak. This limitation isn't fully explored and suggests the attack's universality might have more nuances than presented.
- The evaluation of the model's performance on normal tasks after the attack (to ensure utility is maintained) is somewhat limited in the main paper, relying on 50 questions from the Alpaca dataset.

**Questions:**

n/a

---

### Note · Authors · 2025-12-16

I have read and agree with the venue's withdrawal policy on behalf of myself and my co-authors.